# Analyzing the Differential Impact of Semen Preparation Methods on the Outcomes of Assisted Reproductive Techniques

**DOI:** 10.3390/biomedicines11020467

**Published:** 2023-02-06

**Authors:** Riffat Bibi, Sarwat Jahan, Tayyaba Afsar, Ali Almajwal, Mohamad Eid Hammadeh, Houda Amor, Ali Abusharha, Suhail Razak

**Affiliations:** 1Department of Animal Sciences, Faculty of Biological Sciences, Quaid-i-Azam University Islamabad, Islamabad 45320, Pakistan; 2Department of Community Health Sciences, College of Applied Medical Sciences, King Saud University, Riyadh 11433, Saudi Arabia; 3Department of Obstetrics, Gynecology and Reproductive Medicine, Saarland University Clinic, 66421 Homburg, Germany; 4Department of Optometry, College of Applied Medical Sciences, King Saud University, Riyadh 11433, Saudi Arabia

**Keywords:** magnetic-activated cell sorting, assisted reproductive technique, sperm DNA fragmentation, density gradient centrifugation

## Abstract

Sperm separation plays a critical role in assisted reproductive technology. Based on migration, density gradient centrifugation and filtration, a properly selected sperm could help in increasing assisted reproductive outcomes in teratozoospermia (TZs). The current study aimed to assess the prognostic value of four sperm selection techniques: density gradient centrifugation (DGC), swim-up (SU), DGC-SU and DGC followed by magnetic-activated cell sorting (DGC-MACS). These were evaluated using spermatozoa functional parameters. A total of 385 infertile couples underwent the procedure of intracytoplasmic sperm injection (ICSI), with an isolated teratozoospermia in the male partner. Semen samples were prepared by using one of the mentioned sperm preparation techniques. The improvements in the percentage of normal mature spermatozoa, rate of fertilization, cleavage, pregnancy and the number of live births were assessed. The normal morphology, spermatozoa DNA fragmentation (SDF) and chromatin maturity checked by using chromomycin A3 (CMA3) with DGC-MACS preparation were better compared to the other three methods. Embryo cleavage, clinical pregnancy and implantation were better improved in the DGC-MACS than in the other tested techniques. The DGC-MACS technique helped in the selection of an increased percentage of normal viable and mature sperm with intact chromatin integrity in patients with teratozoospermia.

## 1. Introduction

During the past two decades of advances in infertility treatment, the use of ART emerged as a therapeutic option for managing male and female subfertility [1,2]. However, the current success rates of assisted reproductive technology (ART) remain relatively low [3]. The development of an embryo after the fertilization of oocytes with poor-quality sperm from infertile men is one of the most important factors in intracytoplasmic sperm injection’s (ICSI) success [4,5,6]. Teratozoospermia (TZs) is defined as a percentage of spermatozoa with normal shape under the lesser reference limit. The cutoff values for normality varied greatly in recent decades to 4% [7]. One recent study provided clear evidence that apoptotic alterations are closely correlated to abnormal sperm morphology and DNA damage [8]. Sperm DNA fragmentation (SDF) involves single- or double-stranded (ss or ds) breaks in sperm DNA and can be caused by extrinsic factors (i.e., heat exposure, smoking, environmental pollutants and chemotherapeutics) as well as intrinsic factors (i.e., defective germ cell maturation, abortive apoptosis and oxidative stress [OS]) [9]. SDF has been linked to reduced male fertility potential, reduced fertilization, decreased pregnancy rates, suboptimal embryo quality, increased risk of spontaneous abortions and poor outcomes from assisted reproductive techniques (ART) [10]. The failure of ART processes continues to spark a drive to refine sperm separation procedures and new sperm selection tools to improve the outcome, specifically to improve embryo quality, implantation and pregnancy [6,11,12,13,14].

The primary goal of the semen processing technique is to increase the amount of live, fast-swimming, mature, compacted chromatin and functional sperm capable of normally fertilizing the oocyte. Density gradient centrifugation (DGC) and swim-up (SU) for sperm selection are the most common techniques routinely used for sperm enrichment, with fast swimming and improved morphology that depends on centrifugation and sperm migration [11,15]. These methods rely on selecting sperm with higher motility while ignoring their molecular properties. The effects of the above techniques on invisible anomalies, i.e., phenomena similar to apoptosis and programmed cell death, DNA damage, sperm chromatin, membrane maturation, ultrastructure and the results of ART have not been adequately evaluated [10,16,17,18,19]. Both DCG and swim-up procedures are not designed to effectively select sperm with no DNA damage, intact chromatin and non-apoptotic sperm [20]. Furthermore, the subsequent selection of active sperm in ICSI is the primary selection by an embryologist in terms of sperm motility and best morphology, thereby bypassing natural defensive barriers and leading to the fertilization of oocytes with chromatin-defective sperm [2]. Molecular features of the sperm apoptotic process associated with male reproduction have gained importance in recent years [21]. Numerous studies have demonstrated the importance of systemic cell death (apoptosis) in sperm, which can be associated with lower pregnancy and implantation rates in assisted reproduction [18,22]. The ejaculated human sperm has been shown to exhibit phosphatidylserine (PS) translocation or externalization involved in two processes: (1) in apoptosis and (2) during the fusion of two membranes, i.e., during the process of capacitation and/or at the time of fertilization, which involves an acrosome reaction and sperm/oocyte binding [23,24,25].

In necrotic cells, disruption of membrane integrity results in PS being accessible and serving as a trigger to initiate apoptosis. Magnetic-activated cell sorting (MACS), a sperm selection procedure, simply and effectively uses nanobeads, nanoparticles and magnetic beads that act as conjugation to proteins or antibodies to select sperm. Annexin-V binding has a high affinity to phosphatidylserine (PS) in sperm with impaired membrane integrity, which is one of the reasons for the low fertilization capacity of sperm [25,26,27,28]. This method is practiced along with the traditional methods of sperm selection using gradient DGC and SU methods [29,30]. The use of MACS together with DGC and SU improves sperm viability and maturation and reduces sperm aneuploidy and apoptosis [31]. Contradictory evidence was found on the use of MACS in ART and its success rate [32]. However, few studies found no significant difference between MACS and DGC/SU semen preparation methods for fertility outcomes [20]. This variability can be attributed to large variance with fewer study participants [23,33,34]. Previous literature did not address the affectivity of MACS in infertility with normal and elevated SDF levels in TZs and the relative efficiency of different processing methods for semen preparation to improve sperm DNA quality and chromatin maturity along with ART outcomes in TZs male patients. Based on these inconclusive results about using the MACS approach for ART, an ideal sperm preparation method for sperm selection has yet to be determined.

Using the sperm chromatin dispersion (SCD) assay to measure spermatozoa DNA fragmentation (SDF) and chromomycin A3 (CMA3) to determine chromatin integrity, we evaluated the MACS technique with the classical sperm preparation methods to find the ideal sperm selection method and tried to improve their relationship to build up the quality parameters in TZs. To our knowledge, this is the first study in which four techniques were analyzed, i.e., DGC, SU, DGC-SU and DGC-MACS, to determine the influence of different sperm selection techniques on improving the number of sperm with mature and intact DNA, condensed chromatin and viability in male subjects with teratozoospermia (TZs) and males with elevated (20%) and normal (≤20%) levels of SDF, as determined by a pre-established threshold [35]. This study also further assessed the impact of sperm preparation techniques on ICSI cycle success and the rates of fertilization, fission, pregnancy and implantation.

## 2. Materials and Methods

### 2.1. Study Design and Ethical Clearance

The study was conducted at the Laboratory of Reproductive Physiology, Institute of Zoology, Faculty of Life Sciences, Quaid-i-Azam University Islamabad, Pakistan. Samples were collected from the Fertility & Genetic Services (FGS) and Salma Kafeel Medical Centre (SKMC) in Islamabad, Pakistan. All participants included in the study were informed about the study, and signed informed consent forms were retrieved. Ethical approval to conduct this prospective study was obtained from the Ethics Committee of Salma Kafeel Medical Center Islamabad Pakistan and the Bioethics Committee of the Department of Zoology of Quaid-i-Azam University and Islamabad and assigned Protocol No. BEC-FBS-QAU2016-77.

### 2.2. Participants

Participants included couples undergoing their first ovarian stimulation (who have been unsuccessful in achieving pregnancy after 12 months or more) in which the male partner’s BMI was between 20 and 30 kg/m^2^ from January 2016 to October 2021. 

The inclusion criteria were as follows: age between 18 and 45 years; basic literacy; at least 1-year history of infertility; female partner age <35 years; female partner BMI <24.5–18 kg/m^2^; FSH ≤ 10 IU/L (Day 2 of menstrual cycle); AMH ≥ 1.0 ng/mL; OR antral follicle count > 10; evidence of at least one patent fallopian tube as determined with either a hysterosalpingogram or laparoscopy showing at least one patent fallopian tube or a saline infusion sonogram showing spillage of contrast material; regular cycles defined as ≥25 days and ≤35 days in duration; evidence of ovulation including biphasic basal body temperatures, positive ovulation predictor kits or progesterone level ≥3 ng/mL, couples in which the male partner had a confirmed contributing cause of the couple’s infertility. The exclusion criteria were as follows: patients with recent fever; external genital abnormalities; cryptorchidism; varicocele; presence of anti-sperm antibodies; treatments that may alter spermatogenesis; patients with chronic diseases, e.g., liver/renal disease, hypertension and diabetes; male patient BMI below 19.5 or over 30 kg/m^2^. Teratozoospermia was described according to the WHO 2010 guidelines and Kruger’s strict criteria (sperm morphology less than 4%). The sample size was calculated as previously described [36,37], and the final total number of couples was 385 (Appendix A).

### 2.3. Semen Collection and Analysis

All men produced their semen sample in a sterile labeled jar. The semen sample was produced via masturbation on the day of oocyte aspiration after 2–5 days of abstinence, and the collected semen sample was left to liquefy at 37 °C for 30 min before analysis. Each sample was split into two aliquots. One of these was subjected to analysis for seminal characteristics. Standard semen parameters were assessed according to WHO 2010 standards; to summarize, the sperm number was determined using an upgraded Neubauer chamber after proper dilution. Motility was determined using a Leica microscope DM300 scoring at least 100 spermatozoa/slide, and morphology was determined using Diff-Quik stain (Dade Behring Inc., Newark, NJ, USA). The other aliquot used was for sperm preparation. The final acquired fraction was tested for sperm count and motility, and then it was maintained at 37 °C in the same medium for 15 min until used for inseminating oocytes through ICSI or IVF. Sperm HOS, ROS, sperm DNA fragmentation (SCD) and chromatin maturity (CMA3) were evaluated in selected spermatozoa remaining after oocyte insemination.

#### 2.3.1. Hypo-Osmotic Swelling Test (HOS)

As previously indicated, HOS tests were carried out on neat semen (World Health Organization, 1999) when the sample had completely liquefied. Counting was done after 30 min of incubation at 37 °C. For each patient, 1 mL of semen sample was mixed at room temperature into the 150 mOsm hypo-osmotic swelling solutions. A mixture of sodium citrate (25 mmol/L) and fructose (75 mmol/L) was added to 0.1 mL of semen and left for 30 min at 37 °C. Two hundred spermatozoa were observed with a phase contrast microscope, and the percentage of spermatozoa with tail changes typical of a reaction in the HOS test (swollen, HOS-reactive or HOS-positive spermatozoa) was determined [38].

#### 2.3.2. Reactive Oxygen Species (ROS)

The estimation of reactive oxygen species (ROS) was done using the protocol of [39]. A volume of 0.1 M sodium acetate buffer was prepared by dissolving 4.1 g of sodium acetate in 500 mL of distilled water. The pH was maintained at 4.8. Then, 10 mg of N, N-Diethyl-P-phenylenediamine sulfate salt (DEPPD) was dissolved in 100 mL of sodium acetate buffer, and a second solution was prepared by adding 50 mg of ferrous sulfate (FeSO4) in 10 mL of sodium acetate buffer. Both of the solutions were mixed in a ratio of 1:25 and incubated in the dark for 20 min at room temperature. Then, 20 μL of solution mixture, 1.2 mL of buffer and 20 μL of the sample were taken in a cuvette, and absorbance was checked at 505 nm by using a Smart Spec TM plus Spectrophotometer. Three readings were taken for each sample after every 15 s. A calibration curve was automatically constructed from the slopes, which were calculated based on varying (delta) absorbencies at 505 nm each time (min) and corresponding to the concentration of hydrogen peroxide. ROS levels were calculated by using the analyzer (spectrophotometric plate reader) from the calibration curve and expressed as equivalent to levels of hydrogen peroxide (1 unit = 1.0 mg H_2_O_2_/L). Tests on a sample of 10 normal fertile (NF) men were run as controls.

#### 2.3.3. Sperm DNA Fragmentation (SDF)

The SDF was measured by using a Sperm Nucleus DNA integrity Kit (SCD) (Shenzhen Huakang Biomed Co., Ltd., Shenzhen, China) as reported previously [40] Briefly, 60 μL of each sample was added to an Eppendorf with melted agarose and applied on a glass slide, which was followed by a glass coverslip. The coverslip was placed on ice for 5 min. The samples were processed by acid denaturation for 10 min and then lysed for 19 min. The glass slide was then rinsed for 5 min with distilled water and then sequentially dehydrated for 2 min in 70 percent, 90 percent and 100 percent ethanol baths. A total of 500 sperm were manually calculated after Wright’s staining using bright field microscopy for each slide. To determine the degree of sperm DNA integrity, the dispersion of sperm DNA was calculated. If the SDF was less than 20%, then it was deemed normal.

#### 2.3.4. Chromomycin A3 Staining

The ejaculated spermatozoa and the washed spermatozoa were fixed in methanol/acetic acid 3:1 (4 °C, 5 min), and 50 μL was then spread on slides as a thin film. The slides were then kept in a dark room for 20 min and were then stained with 150 mL of CMA3 (Merck 230752-10MG and Sigma-Aldrich Co, LLC, St. Louis, MO, USA C2659-10MG) (0.25 mg/mL) in McIlvain buffer. Following that, the slides were washed and mounted with a buffer. Under a fluorescent microscope (Olympus, BX41 Japan) with a 460-nm filter, 200 spermatozoa were counted for each sample in duplicate; bright yellow fluorescence-reacted spermatozoa (CMA3+%) were considered chromatin uncondensed, and yellowish-green fluorescence (CMA3) reflected as chromatin condensation. The percentage of sperm with bright yellow-green fluorescence was calculated for each slide. Each slide was evaluated by two individuals. The coefficient variation between the two individuals was less than 10%, and the mean of the duplicate results was used for analysis [7,41,42].

### 2.4. Experimental Design

The semen samples from 385 patients were divided into four groups. Simple randomization using a closed-envelope method was used to randomize matches based on sperm selection techniques; the DGC-MACS separation technique at the SKMC and FGS clinics was used mostly for patients with previous failed ART attempts and with high SDFs. In this way, the population was divided into four groups, i.e., density gradient centrifugation (DGC) *n* = 99, swim-up (SU) *n* = 92, DGC-SU *n* = 100 and DGC followed by magnetic-activated cell selection (DGC-MACS) *n* = 94. These four methods were used in the IVF laboratory where the study was performed. As a standard clinical practice, MACS selection in a clinic is mostly recommended in infertile males with low sperm count, poor sperm morphology and higher SDF > 20%, which is the reason for the basic characteristics of the MACS group’s participants being different from those of the other groups.

#### 2.4.1. Density-Gradient Centrifugation (DGC) Technique

Sperm Grade (Vitrolife, Gothenburg, Sweden) was diluted in medium G-Mpos Plus (Vitrolife, Gothenburg, Sweden) to generate 45 and 90 percent density dilutions for density gradient centrifugation (DGC). In 15 mL Falcon tubes, two 90 percent and 45 percent columns were created by layering 1–1.5 mL of each solution, starting at the bottom with the 90 percent fraction. A volume of 1 mL of the undiluted sample was layered as the top layer of the columns and centrifuged at 300 g for 15 min. The resulting pellet was collected after centrifugation and washed once at 350 g for 10 min.

#### 2.4.2. Swim-Up (SU) Technique

For the sperm swimming method, the semen sample was used without centrifugation. A sample of 0.5–1.0 mL was carefully layered under 0.5 mL G-Mops plus (Vitrolife, Gothenburg, Sweden) and incubated for 1 h at 37 °C. The final fraction contained only the top 0.25 mL fraction, which was carefully collected into a new tube.

#### 2.4.3. DGC-SU Technique

For the DGC-SU procedure, the sperm sample was prepared first with the DGC technique as mentioned in the above section. The final 0.25 mL pellet was layered gently under 0.5 mL of G-Mops plus (Vitrolife, Gothenburg, Sweden) and incubated for 1 h at 37 °C. The final fraction included only the topmost 0.25 mL fraction, which was collected gently into a new tube.

#### 2.4.4. DGC-MACS Technique

Samples were subjected to a non-apoptotic selection technique using the MACS ART Annexin V Reagent (Madison, CT, USA) and according to the manufacturer’s instructions. Cells obtained after density gradient centrifugation were centrifuged at 300× *g* for 5 min. The cells were re-suspended in a 90 mL binding buffer after removing the supernatant. The sperm suspension was then incubated for a further 20 min and then transferred to another column where, after washing it with 2 mL of binding buffer (Madison, CT, USA), it was bound to the magnet. The apoptotic sperm were kept in the separation column, and the non-apoptotic sperms in the negative fractions were collected in a tube after passing through the column. Finally, after discarding the apoptotic sperm fraction, the non-apoptotic sperm fraction was centrifuged. Each prepared fraction was divided into 2 aliquots and used for the analysis of sperm parameters, vitality, reactive oxygen species and sperm DNA damage as chromatin condensation after one fraction was used in ICSI.

#### 2.4.5. Ovarian Stimulation, IVF, ICSI and Embryo Development

Following a long protocol, mid-luteal phase long-acting gonadotropin-releasing hormone analogues (triptorelin, Decapeptly, France, Ipsen Pharma) followed by an exogenous individual dose of recombinant follicle stimulation hormone r-FSH (Gonal-F, Merk Serono- Germany) were used to induce multiple follicular growths, with starting doses ranging from 150 to 225 IU according to age, body mass index, antral follicular count, AMH level and response to previous stimulation. The stimulation concentration’s titer was measured according to ovarian response (estradiol level and ultrasound every 2 days until at least two follicles reached 17 mm in diameter. Finally, at 34–36 h following the delivery of human chorionic gonadotrophin u-HCG (IVF-C, LG Lifesciences), oocytes were harvested transvaginally with ultrasound guidance from patients sedated with general anesthesia, and the oocytes were cultured in human tubal fluid supplemented with 5% human serum albumin (HSA) in a 5% CO_2_ humidified gas environment at 37 °C. Depending on sperm indices and the couples’ reproductive histories, the oocytes were inseminated using conventional IVF, and cumulus oocytes were incubated with 60,000 spermatozoa/oocyte in in vitro fertilization supplemented with HSA-IVF-plus medium (Vitrolife, Goteborg, Sweden) or, for ICSI, an OLYMPUS IX51/71/81/53/73/83 microscope assembled with INTEGRA Ti microinjector was used. The oocytes were assessed at 16–18 h after insemination based on the presence of two pronuclei. Individually fertilized oocytes were sequentially cultivated in G1/G2 Plus (Vitrolife, Goteborg, Sweden), incubated in an MIRI multiroom incubator (Singapore, Esco Medical) and scored 40, 62, 88 and 112 h after insemination. The number and shape of nuclei and blastomeres were counted as well as the percentage and kind of fragmentation [5]. At 62–64 h after insemination, developing embryos were monitored for grade and stage, and then, after 112 h, patients were called for embryo transfer. Patients followed up with a pregnancy test and with ultrasound for a positive heartbeat.

The study protocol was conducted following the principles of the Declaration of Helsinki [43].

### 2.5. Statistical Analysis

For the statistical analyses, we used the Statistical Package for the Social Sciences (IBM SPSS software, version 20). All parameters had a normal distribution. For the descriptive analysis of the results, the data are expressed as mean ± standard deviation (SD). Differences between means were evaluated using ANOVA (*p*-value < 0.05), and groups were compared with Tukey’s test. The Wilcoxon signed-rank test was used to access the significance of differences post-preparation, and the nonparametric Friedman test for paired samples was applied to assess the efficacy of the preparation group over others. The reliability of the prediction produced by the model was statistically tested by the Hosmer–Lemeshow goodness-of-fit test.

## 3. Results

### 3.1. Demographic Characteristics

There was no significant difference in the demographic characteristics (mean male patient age, body mass index (BMI), duration of infertility, female patient age and Anti Mullerian hormone (AMH)) of all four groups (DGC, SU, DGC-SU and DGC-MACS) (Table 1).

### 3.2. Sperm Standard and Quality Parameters

Sperm count, normal morphology, total motile sperms (TNMS) in the neat ejaculated semen, the levels of white blood cells (WBC), reactive oxygen species (ROS), sperm parameters viability (HOS %), percent sperm DNA fragmentation (SDF) levels and percent CMA3+ % levels were comparable between all study groups (*p* > 0.05) (Table 2).

### 3.3. Post-Sperm Selection Technique Improvement in Standard and Quality Parameters of Sperm

Assessment of the efficacy of DGC-MACS in improving semen parameters post-preparation was compared with all three preparation techniques (DGC, SU, DGC-SU) in improving the mean percentages of sperm HOS %, SDF %, CMA3+ % and ROS (units), as shown in Table 3 and Table 4. There was a non-significant improvement in HOS % and a significant (*p* < 0.00) decrease in ROS% (*p* < 0.00), SDF % (*p* < 0.00) and CMA3+ % (*p* < 0.00) in DGC-MACS-prepared sperms when compared with the other sperm preparation techniques, as shown in Table 3. A significant (*p* < 0.01) improvement in percent total normal motile sperm (TNMS) after DGC-MACS (76 ± 26%) was obtained compared to DGC (67 ± 23%), SU (68 ± 28%) and DGC-SU (64 ± 26%) preparations. The CMA3+ % level (immature sperm concentration) after preparation in DGC-MACS was 21.9 ± 7 and significantly (*p* = 0.00) lower compared to that of DGC (27.5 ± 10), SU (28.1 ± 11) and DGC-SU (27.3 ± 10.4)(Table 4).

### 3.4. Effect of Sperm Preparation Techniques on Quality Sperm Selection in Men with Normal and Increased Sperm DNA Fragmentation

The neat semen samples we compared with prepared sperm in all four groups, separated further into two cohorts of SDF ≤20 and >20%, showed significant (*p* < 0.01) improvement in mature sperm selection (CMA3+%) post-preparation when compared with the pre-preparation measurements. This improvement was greater for DGC-MACS in both the SDF ≤20% (z = −4.92, *p* < 0.00) and SDF >20% (z = −6.4, *p* < 0.00) cohorts compared to all other preparation techniques. Similarly, CMA3+ % values in the SDF levels ≤20% cohort treated with DGC-MACS showed 18.4 ± 5 post-preparation; the DGC group showed 30.53 ± 10.1; the SU group showed 27.7 ± 12; the DGC-SU group showed 30.1 ± 10.6. CMA3+ % levels after DGC-MACS preparation were statistically significantly (*p* = 0.00) lower in the SDF ≤ 20% cohort, whereas in the SDF >20% cohort, the CMA3+ % levels were as follows: DGC-MACS, 24.5 ± 8; DGC, 28.9 ± 10.8; SU, 28.4 ± 10.4; DGC-SU; 26.06 ± 9.98. In these groups, we found no significant difference between pre- and post-preparation (*p* = 0.06). It is clear from the box-and-whisker plot in Figure 1 that the DGC-MACS group showed a significant reduction in the population of chromatin de-condensed (CMA3+ %) spermatozoa or immature sperm in samples with ≤20% SDF.

### 3.5. Effect of Sperm Selection Techniques on Assisted Reproductive Technology Cycle Parameters

The ART cycle parameters and outcome of sperm preparation (DGC, SU, DGC-SU and DGC-MACS) techniques are reported in Figure 2. There was a significant (*p* < 0.05) increase in cleavage rate in the DGC MACS group. All other parameters are comparable between all preparation groups.

### 3.6. Effect of Sperm Selection Techniques on Assisted Reproductive Technology Cycle Parameters in Men with Normal and Increased Sperm DNA Fragmentation

The DGC-MACS group had significantly (*p* < 0.01) improved cleavage rates in both SDF was ≤20% and >20% cohorts, whereas there was no difference in the live implantation rate in both SDF cohorts (Table 5, Figure 3).

### 3.7. Effect of Sperm Selection Techniques on Assisted Reproductive Technology Cycle Outcomes

Pregnancy rate (DGC-MACS, 52.5 ± 45; DGC, 44.4 ± 49; SU, 35.8 ± 48; DGC-SU, 36 ± 48) was significantly (*p* < 0.01) higher in the DGC-MACS group, whereas the implantation rate was comparable (*p* > 0.05) in all preparation groups (Table 5).

## 4. Discussion

It has been reported that ejaculated spermatozoa do exhibit changes consistent with apoptosis in somatic cells, such as being external of PS, disrupted mitochondrial membrane potential and/or DNA fragmentation. The data confirmed the presence of apoptotic markers (EPS) in spermatozoa prepared by DGC for either healthy donors or subfertile couples [44]. It has been reported that sperm motility decreased after MACS compared with spermatozoa after DGC because of the further centrifugation steps in patients with compromised motility and count [45]. The data in the current study show better selection of sperm using the DGC-MACS technique. Concerning a better selection of sperm with normal morphology and motility, the molecular content of DGC-MACS-selected sperm samples revealed mature spermatozoa of condensed chromatin with lower SDF rates along with lower ROS content and better vitality (HOS). These results suggest that sperm function may not be affected by further manipulation during MACS. A study previously done showed a strong correlation between sperm DNA fragmentation and poor sperm quality, although no preferential effect on sperm concentration or morphology seemed to be present [46,47]. Recently, apoptosis received much attention because of its vital role in reproduction, and phosphatidylserine, which is normally sequestered in the plasma membrane inner leaflet, appears in the outer leaflet where it triggers non-inflammatory phagocytic recognition of the apoptotic cell [27,48,49]. Magnetic-activated cell sorting (MACS) is a new method for selecting spermatozoa that has the advantages of simplicity, low cost, specificity and sensitivity [50]. Magnetic cell sorting can label and separate PS-translocated sperm because membrane integration is disrupted at the molecular level, which is an apoptotic symptom [25,27,45,51].

There are very limited studies evaluating whether DGC-MACS improves reproductive outcomes, which are determined in terms of the number of oocytes, the embryos and the transfer procedures used to achieve parturition [33,47]. Our results indicate that adopting the DGC-MACS approach to human sperm preparation can result in increased division and chemical pregnancy rates as well as a trend towards improved implantation rates [27,52]. In the present study, we divided the male groups into a normal SDF and higher SDF range cohort to assess the true benefit of DGC-MACS. We have not observed a significant advantage in choosing DGC-MACS for either cohort of SDF with a 20% cutoff, and similar findings were reported before [35]. DGC-MACS sperm selection is currently offered to patients in very specific cases involving males with elevated SDF or numerous failed ART attempts with no obvious female partner-originating cause and without using a preferred method [16,27,28,32,53,54,55]. Considering the controversy surrounding the introduction of add-ons into clinical practice without proper assurance and thorough reviews, clinicians must have reliable records as well as use appropriate statistical methods, designs and unbiased data to ensure patients receive fertility treatment options tailored to their needs and preferences [29,32,34].

Conventional flotation or one-step washing should not be used to separate pure sperm with increased oxidative stress due to infection or the risk of collecting immature and damaged sperm [56]. The cell contents of the ejaculate are driven to the pellet, allowing infected and defective sperm to mix with mature sperm and cause damage [2,34]. DGC shows promise for isolating mature, infection-free sperm, whereas DGC-MACS has faster and easier sorting technology at high concentrations [11,31]. When defective sperm are removed, oxidative stress is reduced in important and healthy cells [57]. As a result, less oxidative stress in the ejaculate after DGC-MACS selection could be another possible explanation for the increased cleavage rates found after DGC-MACS separation in this study [27,52]. The present study demonstrated the influence of magnetic passage on reducing oxidative stress as reported before [23,45,51,58]. The results in the current study indicate that an increased cleavage rate as well as a trend towards better pregnancy and implantation could be due to the removal of genetically defective cells containing apoptotic properties [27]. Thus, the DGC-MACS separation of non-apoptotic spermatozoa leads to selection for balanced gametes and improved result rates in TZs. There were no significant differences in the implantation rate across all groups and cohorts, but a non-significant improvement was observed in the DGC-MACS group [31,51,59,60]. It should be noted that the number of patients studied in previous data, including the current study, is small, and most studies are underpowered to determine ART outcomes. Further studies are needed to better understand this process and determine the true value of this technique. Before sophisticated sperm selection procedures are used extensively in ART, the safety and efficacy of the procedure should be thoroughly evaluated.

## 5. Conclusions

In conclusion, the findings of the present study show that the DGC-MACS preparation technique is a method of choice to improve the percentage of mature, normal and viable sperm with condensed chromatin and intact genetic integrity in patients with teratozoospermia, and it shows that it is safe and improves assisted reproduction outcomes. Further research involving a larger TZs patient population and more applications could yield more comprehensive information about the ART results.

## Figures and Tables

**Figure 1 biomedicines-11-00467-f001:**
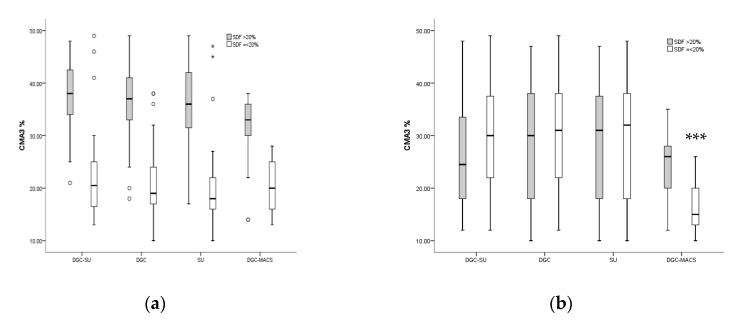
Values of CMA3+ % (**a**) from neat semen samples and (**b**) measured after spermatozoa selection using DGC-SU, DGC, SU and DGC-MACS. Semen samples separated here by ≤20 and >20% SDF. ** *p* < 0.01, *** *p* < 0.001.

**Figure 2 biomedicines-11-00467-f002:**
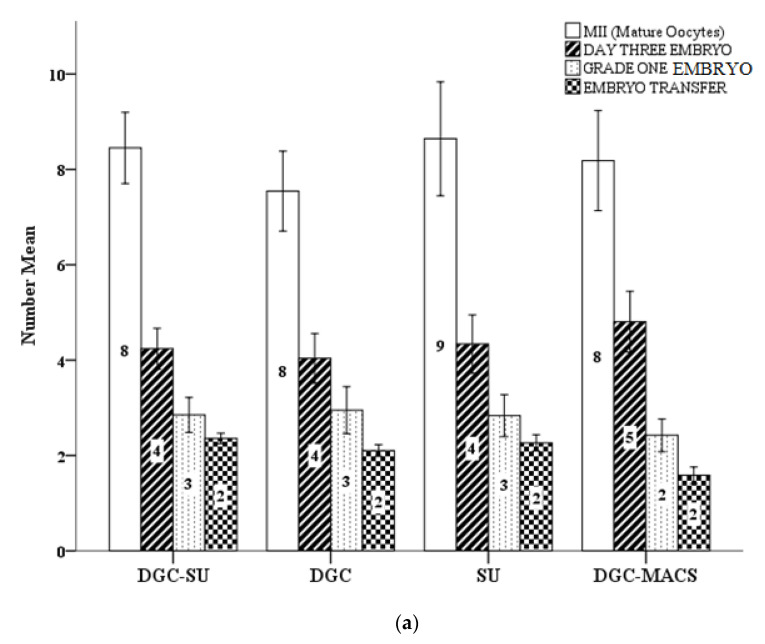
Effect of different semen preparation methods on assisted conception outcomes. (**a**) Number of MIIs (mature oocytes), number of day three embryos, number of grade one embryos and number of embryos transferred; (**b**) percentage of mature oocytes, fertilization rate, cleavage rate and blastocyst rate.

**Figure 3 biomedicines-11-00467-f003:**
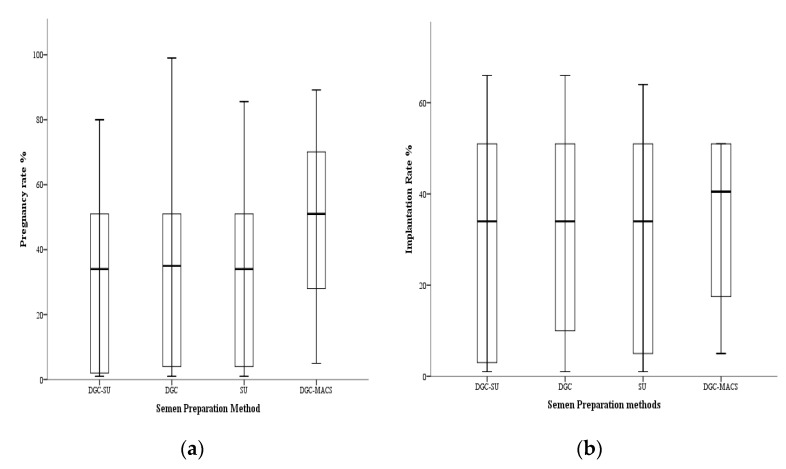
Box plot showing clinical (**a**) pregnancy rate and (**b**) implantation rate in couples in which semen preparation was done using the DGC-SU, DGC, SU and DGC-MACS methods.

**Table 1 biomedicines-11-00467-t001:** Male patient age, male patient body mass index (BMI), female patient age and anti-Mullerian hormone (AMH) in the whole studied population of patients with TZs.

	DGC-SU (N = 100)	DGC (N = 99)	SU (N = 92)	DGC-MACS (N = 94)	*p*-Value
Male patient age (Years)	34.89 ± 3.9	34.90 ± 4.50	34.9 ± 646	39.07 ± 8.59	0.93
Male patient BMI (Kg/m^2^)	25.34 ± 3.46	25.25 ± 3.1	24.5 ± 3.02	25.39 ± 2.98	0.09
Infertility duration (Years)	9.32 ± 5.38	9.54 ± 5.57	9.24 ± 6.42	10.34 ± 6.16	0.19
Female patient age (Years)	30.7 ± 5.31	30.86 ± 5.61	31.8 ± 6.08	33.01 ± 5.42	0.28
Female patient BMI (Kg/m^2^)	27.10 ± 4.52	28.95 ± 2.58	28.78 ± 3.66	28.14 ± 3.81	0.49
AMH (ng/mL)	3.98 ± 2.94	4 ± 3.70	3.30 ± 3.30	3.6 ± 3.5	0.56
Total gonadotropin dose (IU)	2529 ± 1536	2539 ± 1878	2925 ± 1755	2625 ± 1834	0.92
Total gonadotropin dose/oocyte	647.6 ± 891	605 ± 1079	747 ± 1011	651 ± 987	0.72
Stimulation days	15 ± 3	15 ± 2	14.5 ± 2.8	15 ± 2	0.89
Estradiol level on the day of HCG (pg/mL)	1974 ± 1316	1791 ± 1076	2040 ± 1408	1804 ± 1601	0.38
Estradiol level/oocyte	237 ± 385	220 ± 302	202 ± 304	200 ± 392	0.586

Values represent mean ± SD.

**Table 2 biomedicines-11-00467-t002:** Sperm concentration, volume, WBC/HPG, TNMS (total normal motile sperms), normal % (normal morphology), HOS % (hypo-osmotic swelling), SDF% (sperm DNA fragmentation) and CMA3% levels in the whole studied population.

	DGC-SU (N = 100)	DGC (N = 99)	SU (N = 92)	DGC-MACS (N = 94)	*p*-Value
Sperm concentration M/mL	22.39 ± 2.76	27.33 ± 9.18	40.00 ± 23.9	30.17 ± 9.44	0.26
Volume (mL)	3.65 ± 1.59	3.95 ± 1.97	3.93 ± 1.52	3.63 ± 1.65	0.33
WBC/HPF	2.89 ± 1.96	2.91 ± 1.79	3.16 ± 1.78	4.49 ± 7.45	0.78
TNMS %	32.25 ± 20.7	43.9 ± 22.5	44.7 ± 22.65	37.40 ± 22.13	0.07
Normal %	2.8 ± 0.68	2.58 ± 0.65	2.4 ± 0.81	2.70 ± 0.93	0.36
HOS %	39.25 ± 27.13	49.09 ± 25.31	51.13 ± 25.9	54.65 ± 19.90	0.79
SDF %	25.15 ± 11.8	20.9 ± 13.2	23.1 ± 12.6	25.3 ± 11.7	0.68
CMA3+ %	27.5 ± 10.3	28.7 ± 10.7	29.3 ± 103	29.6 ± 12.4	0.45

Values represent mean ± SD.

**Table 3 biomedicines-11-00467-t003:** TNMS (total normal motile sperms), HOS % (hypo-osmotic swelling), SDF% (sperm DNA fragmentation) and CMA3+% levels in the whole studied population after preparation of patients with TZs.

	DGC-SU (N = 100)	DGC (N = 99)	SU (N = 92)	DGC-MACS (N = 94)	*p*-Value
HOS %	65.8 ± 25.8	66.0 ± 23.3	65.1 ± 24.7	73.8 ± 17.5	0.253
SDF %	14.2 ± 3.5	14.7 ± 4.7	14.5 ± 3.9	12.3 ± 4.7 ^bc^	0.01
CMA3+ %	27.5 ± 10.1	29.7 ± 10.48	28.1 ± 11.1	21.9 ± 7.4 ^abc^	0.001

Values represent mean ± SD. ^a^ = DGC-SU vs. DGC, SU, DGC-MACS; ^b^ = DGC vs. SU, DGC-MACS; ^c^ = SU vs. DGC-MACS.

**Table 4 biomedicines-11-00467-t004:** ROS (units) (reactive oxygen species) levels in all studied subjects before and after preparation of semen of normal fertile (NF) men (control) and men with teratozoospermia (TZs).

	NF (Control)	Teratozoospermia (TZs)				
		DGC-SU	DGC	SU	DGC-MACS	*p*-Value
Neat sample	1.8 ± 0.2 (N = 10)	23.86 ± 1.4 (N = 100)	29.7 ± 1.91 (N = 99)	29.78 ± 1.18 (N = 92)	28.95 ± 1.10 (N = 94)	0.9
Post-preparation	0.4 ± 0.1 (N = 10)	1.4 ± 1.0 (N = 100)	1.1 ± 1.02 (N = 99)	1.1 ± 1.0 (N = 92)	0.53 ± 0.5 ^abc^ (N = 94)	0.001

Values represent mean ± SD. ^a^ = DGC-SU vs. DGC, SU, DGC-MACS; ^b^ = DGC vs. SU, DGC-MACS; ^c^ = SU vs. DGC-MACS.

**Table 5 biomedicines-11-00467-t005:** MII (oocyte metaphase two), 2PN (two pronuclei), fertilization rate, D3 EMB (day three embryo), cleavage rate and number (N) of embryos transferred in the four processing group populations with < 20 and ≥20% SDF in each studied population.

SDF ≤ 20%						SDF > 20%				
	DGC (36)	SU (42)	DGC-SU (32)	DGC-MASC (39)	*p*-Value	DGC (64)	SU (50)	DGC-SU (55)	DGC-MASC (55)	*p*-Value
MII	8.2 + 6.1	8.1 + 6.3	7.1 + 4.8	8 + 5	0.83	7.3 + 7.1	9 + 5.6	8.5 + 56	8.6 + 5.8	0.74
2PN (Fertilization rate %)	5 + 4.4 (66)	5.7 + 4.5 (73)	5.1 + 4.3 (69)	5.7 + 4 (70)	0.49	5.1 + 4.8 (73)	6.1 + 4.4 (74)	5.3 + 3.9 (68)	5.8 + 4.2 (58)	0.66
D3 EMB (Cleavage rate %)	3.4 + 3.1 (79)	4.4 + 3.6 (81)	4.3 + 4.02 (85)	5.1 + 3.4 (93)	0.25	4.08 + 4.4 (85)	4.0 + 3.1 (75)	3.6 + 2.6 (77)	4.5 + 2.7 (87) ^c^	0.03
Number of Embryos transferred	2.3 ± 0.7	2.4 ± 0.7	2.4 ± 0.88	1.4 ± 0.75	0.66	1.8 ± 0.7	2.3 ± 0.76	2.1 ± 0.86	1.6 ± 0.8 ^abc^	0.00
implantation rate %	21 ± 42.5	35.7 ± 49	19 ± 40	36 ± 45	0.19	35 ± 48	28 ± 46	34 ± 48	38 ± 42	0.92

Values represent mean ± SD. ^a^ = DGC vs. SU, DGC-SU, DGC-MASC; ^b^ = SU vs. DGC-SU, DGC-MASC; ^c^ = DGC-SU vs. DGC-MASC.

## Data Availability

All data generated or analyzed during this study are included in this published article. All of the data are contained in the manuscript. The corresponding author should be contacted if someone wants to request the data from this study.

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
