# Peer review of "Analyzing the Differential Impact of Semen Preparation Methods on the Outcomes of Assisted Reproductive Techniques"

_biomedicines, 2023, doi:10.3390/biomedicines11020467_

Round 1

Reviewer 1 Report

Bibi et al. compared different preparation of sperm suspensions from 385 patients on sperm concentration, motility, morphology, DNA integrity, and level of protamination and determined which preparation allows higher percentages of embryo cleavage, clinical pregnancy, and implantation. They found that the DGC-MACS technique selects normal viable spermatozoa with less DNA damage and higher protamination levels than the others.

Specific comments:

The manuscript was difficult to read as it contains a significant number of typos and problems with English grammar.

The material and methods section lacks important information. There is no indication of whether female infertility was ruled out in these couples. Methods are incomplete and briefly explained. It is well known that the CMA3 technique is more accurate when using flow cytometry (testing ~20,000 spermatozoa) compared to the microscopic technique used here (testing at least 100 spermatozoa, as the authors indicated). The sperm DNA fragmentation method used is less sensitive than the SCSA, which is now the gold standard for this type of damage. Other parameters indicating apoptotic-like changes (e.g. caspases activation, cytochrome C release, etc.) were not determined.

The discussion is a mere repetition of results with some inaccuracies. For instance, apoptotic-like changes can occur during spermatogenesis in germ cells, during epididymal maturation, or after ejaculation.

It is well known that selection using DGC and MACS techniques allows the selection of spermatozoa with better quality than the raw sample, thus increasing the potential success rate of ICSI. The differences among the different preparation methods are not big enough to incline DGC-MACS over the others since the fertilization and implantation rates are similar.

Author Response

Respected Editor and Reviewer

I am highly thankful and obliged to the respected reviewers and editors for the valuable time that you invested in re-evaluating the paper and thanks for the valuable comments you have provided. We have addressed all the comments raised by reviewers. Following are the answers

Reviewer1

The manuscript was difficult to read as it contains a significant number of typos and problems with English grammar.

Response: All grammatical errors have been removed

material and methods

The material and methods section lacks important information. There is no indication of whether female infertility was ruled out in these couples.

Response: A statement added in the material methods defining whether female infertility was ruled out in these couples

“couples with male factor with confirmed contributing cause of the couple’s infertility and female infertility was ruled out in these couples”.Methods are incomplete and briefly explained. It is well known that the CMA3 technique is more accurate when using flow cytometry (testing ~20,000 spermatozoa) compared to the microscopic technique used here (testing at least 100 spermatozoa, as the authors indicated).

Response: Confocal fluorescence microscopy is more advantageous or convenient to flow cytometry in terms of simplicity and required laboratory time, since the sample does not require pre-treatment. In addition, it provides information on cell morphology.

Actually we counted 500cells but at-least 100 for each sample.

 “500 spermatozoa were observed and at least 100 spermatozoa were counted for each sample; bright yellow fluorescence reacted spermatozoa (CMA3+) were considered chromatin uncondensed, while yellowish-green fluorescence (CMA3) were reflected as chromatin condensation [1, 2]”.

The sperm DNA fragmentation method used is less sensitive than the SCSA, which is now the gold standard for this type of damage. Other parameters indicating apoptotic-like changes (e.g. caspases activation, cytochrome C release, etc.) were not determined.

Response: As it was a clinical study and we had evaluated the techniques based on clinical scenario so we find flow cytometer utility in clinical setup requires the presence of expensive instrumentation (flow cytometer) and highly skilled technicians [3].

The discussion is a mere repetition of results with some inaccuracies. For instance, apoptotic-like changes can occur during spermatogenesis in germ cells, during epididymal maturation, or after ejaculation.

Response: Rewritten according to suggestion  

“Magnetic activated cell sorting (MACS) is a new method for selecting spermatozoa that has the advantages of simplicity, low cost, specificity, and sensitivity [4]. Magnetic cell sorting can label and separate PS-translocated sperm because membrane integration is disrupted at the molecular level, which is an apoptotic symptom [5-8]. “

It is well known that selection using DGC and MACS techniques allows the selection of spermatozoa with better quality than the raw sample, thus increasing the potential success rate of ICSI. The differences among the different preparation methods are not big enough to incline DGC-MACS over the others since the fertilization and implantation rates are similar.

Response: Agreed with the statement but “a non-significant improvement in fertilization and implantation rate was observed in DGC-MACS group” secondly the DGC-MACS sperm selection is currently offered to patients in very specific cases and without using a preferred method (in clinic from where the samples were collected), in cases involving males with elevated SDF or numerous failed ART attempts with no obvious female cause.

Reviewer 2 Report

In the current study the authors aimed to assess the prognostic value of four sperm selection techniques: i.e., density gradient centrifugation (DGC), Swim-up (SU), DGC-SU, or DGC followed by magnetic-activated sperm selection (DGC-MACS), on spermatozoa functional parameters. The manuscript is clear and well written. However, some issues should be addressed before considering for publication.

1. The authors should clarify the meaning of the abreviations cited for the first time in the introduction (abstract exluded) to avoid confusion or misunderstanding

2. Please clarify the identification of the four study group and the patients allocation: were the patients randomized? It is not clearly mentioned that this is a prospective study and if the four methods are currently used in the IVF lab where the study was performed

3. The authors should further discuss the findings in relation to a cost-effectiveness evaluation to perform in the daily routine of an IVF lab

Author Response

Respected Editors  and Reviewer

I am highly thankful and obliged to the respected reviewers and editors for the valuable time that you invested in re-evaluating the paper and thanks for the valuable comments you have provided. We have addressed all the comments raised by reviewers. Following are the answers

Reviewer 2

In the current study the authors aimed to assess the prognostic value of four sperm selection techniques: i.e., density gradient centrifugation (DGC), Swim-up (SU), DGC-SU, or DGC followed by magnetic-activated sperm selection (DGC-MACS), on spermatozoa functional parameters. The manuscript is clear and well written.

Thanks for encouragement

 However, some issues should be addressed before considering for publication.

  1. The authors should clarify the meaning of the abbreviations cited for the first time in the introduction (abstract exluded) to avoid confusion or misunderstanding

Response: Done as per suggestion

  1. Please clarify the identification of the four study group and the patient’s allocation: were the patients randomized? It is not clearly mentioned that this is a prospective study and if the four methods are currently used in the IVF lab where the study was performed

Response: Simple randomization using a closed envelop method was used to randomize matched based for sperm selection techniques, (for DGC-MACS separation technique at SKMC & FGS clinic used it mostly for patients with previous failed ART attempt and with high SDF),

  1. The authors should further discuss the findings in relation to a cost-effectiveness evaluation to perform in the daily routine of an IVF lab

Response: Added to text as suggested. “Magnetic activated cell sorting (MACS) is a new method for selecting spermatozoa that has the advantages of simplicity, low cost, specificity, and sensitivity [4].”

Reviewer 3 Report

1.     Define SDF in the introduction

2.     In the introduction, please introduce the teratozoospermia and define its pathological mechanism. Cit new articles regarding teratozoospermia (check the articles of Ammar et al.)

3.     Please put evidence on oxidative stress, apoptosis and SDF.  

4.     In MM, how it is possible 35 days of abstinence.

5.     All P should be in italic and lower case

6.     the Methods please clarify (if known) whether males included in this study were clinically identified as potentially infertile, or the reason these couples were referred to the infertility clinic lies with their female partner.

7.     Can you please clarify the type of teratozoospermia

8.     Can you please detail the used method to classify teratozoospermia.

9.     Can you please add the spermocytogramm of the studied groups.

10.  I suggest also to check statistical correlations  

11.  In the discussion, authors are comparing the used techniques to select spermatozoa however they did not discuss the clinical evidence of their results. Moreover, the pathway how these methods induce sperm damage at different levels of OS, SDF and apoptosis is not well discussed. I recommend to rewrite the discussion.

12.  In general English spelling and grammar need revision.

Author Response

Respected Editor and Reviewer

I am highly thankful and obliged to the respected reviewers and editors for the valuable time that you invested in re-evaluating the paper and thanks for the valuable comments you have provided. We have addressed all the comments raised by reviewers. Following are the answers

Reviewer 3

  1. Define SDF in the introduction

Response: Added in article “Sperm DNA fragmentation (SDF) involves sperm DNA single- or double-stranded (ss or ds) breaks, can be caused by extrinsic factors (i.e., heat exposure, smoking, environmental pollutants, and chemotherapeutics) as well as intrinsic factors (i.e., defective germ cell maturation, abortive apoptosis, and oxidative stress [OS]) [9].

  1. In the introduction, please introduce the teratozoospermia and define its pathological mechanism. Cit new articles regarding teratozoospermia (check the articles of Ammar et al.)

Response: Done as suggested

“Teratozoospermia (TZs) is defined as a percentage of spermatozoa with normal shape under the lesser reference limit.  The  cutoff  values  for  normality  varied  greatly  in  recent  decades to 4% [10]”.

  1. Please put evidence on oxidative stress, apoptosis and SDF.

Response:  Done and included

  1. In MM, how it is possible 35 days of abstinence.

Response: It was a typing error “All men produced their semen sample in a sterile labelled jar, the semen sample was collected by masturbation on the day of oocyte aspiration, after 2–5 days of abstinence,”

  1. All P should be in italic and lower case

Response: Changes made as suggested

  1. the Methods please clarify (if known) whether males included in this study were clinically identified as potentially infertile, or the reason these couples were referred to the infertility clinic lies with their female partner.

Response: Statement included in text “couples with male factor with confirmed contributing cause of the couple’s infertility and female infertility was ruled out in these couples.”

  1. Can you please clarify the type of teratozoospermia

Response: Done already

  1. Can you please detail the used method to classify teratozoospermia.

Response: Teratozoospermia was described according to the WHO 2010 guidelines and Kruger’s strict criteria (sperm morphology less than 4%).

  1. Can you please add the spermocytogramm of the studied groups?

Response:  Added tTable 2. Sperm concentration, volume, WBC/HPG, TNMS (total normal motile sperms), Normal % (normal morphology), ROS% (reactive oxygen species), HOS % (Hypo osmatic test), SDF% (sperm DNA fragmentation), and CMA3% levels in the whole studied population.

  1. I suggest also to check statistical correlations  

Response: Done

  1. In the discussion, authors are comparing the used techniques to select spermatozoa however they did not discuss the clinical evidence of their results. Moreover, the pathway how these methods induce sperm damage at different levels of OS, SDF and apoptosis is not well discussed. I recommend to rewrite the discussion.

Response: Discussion rewritten according to suggestion.

  1. In general English spelling and grammar need revision.

Response: All Grammatical errors have been removed

Reviewer 4 Report

Overall, the manuscript needs English improvement. The manuscript must be carefully revised by the authors as it presents some mistakes and typographical errors that need to be corrected.

Other than that, there are other points that need author’s attention and these are listed below.

Abbreviations should be explained within parentheses at their first mention in the manuscript.

Introduction, L 20/21, “However, the current success rates of ART remain subnormal”: What does it mean “subnormal”? Did you mean "relatively low" success rates?

Materials and Methods section needs to be reorganized and duplicities should be omitted (L 95/96 and L 115/116, L 94/95, L 109/111 and L 116, etc.). The study design is not clear. If the semen analyses were performed to select samples/participants for semen preparation methods, that should be clearly mentioned. Add "Sample collection and preparation" section.

Study design should include study subjects to be investigated (Inclusion/Exclusion criteria) with number of participants, Ethical issue, study groups.

2.2. Participants: First, male participants should be described, than their female partners. This section needs some revision to be more clear. For example, L95/96, “no infertility factor in the couple’s partner were included in the study”. What does it mean? Does it mean that male factor is a (confirmed)  contributing cause of the couple infertility? L 99/100: some of the andrological disorders are mentioned in the L 97.

2.3. This section is not clear. Is it related to study power calculation? L 106-111: this should be moved elsewhere. 

2.4. Please explain the choice of 35 days of abstinence? Isn’t it too long (usually (and in WHO laboratory manual) it is suggested to abstain for 2-7 days before human semen sample collection). 

2.4.1. In how many total spermatozoa cells counting was performed (to calculate the mean percentage)?

L 121-124: It is not clear – split the sentence. 

2.4.2. ROS: Please, add the units in which the ROS value is expressed? What was the control? 

2.4.4. How is the semen thin film prepared? Who is producer of CMA3 staining? What is a model of a fluorescence microscope? Please, provide Brand Name, Place of manufacture (City, Country).   

Table 1: Comparative data for pre-pregnancy female BMI is missing. Statistical test used for data comparison should be added. Units for Anti Mullerian hormone levels, Total Gonadotropin dose per oocyte and Estradiol level are missing and should be added. 

Table 2, sperm concentration: What does it mean mx10exp6? Sperm count (total number of sperm in a sample of one ejaculate) is usually in the ranges of millions (and should be expressed as (UPPERCASE) M or 10exp6/ejaculate), and sperm concentration refers to the number of sperm per unit of volume (mL) of semen and is expressed in millions per mL (M/mL or 10exp6/mL). ROS%: what is used for the control? 

Discussion, 3.4. L 266, “a cohort of SDF with < 20 and ≥ 20%”, L 270 “cohort with SDF levels ≤ 20 %”, L 272 “in SDF ≥ 20 group”: If one group has SDF ≥ 20%, another group should have SDF < 20. 

L 268/269, “This improvement was more after DGC-MACS (z=-4.92, p<0.00) and (z=-6.4, p<0.00).”: It is not clear (for which group is first z and p, for which is the second). Please, revise all section to be clear what is significantly higher/lower in comparison to what.   

Figure 1: If a difference in CMA3 % for some selection methods between the SDF groups is statistically significant (as is mentioned in the text (section 3.4.), please add * above the corresponding Box & Whisker. Explanation for Boxes and Whisker levels is missing (How are data spread out? What represents boxes and lines, what whiskers?). Unit for CMA3 is missing. 

Figure 1, Figure 2: Both figures show CMA3% levels for two SDF groups depending on the semen preparation methods. It is not clear. 

Discussion: It is not clear why is reference cited “We have not observed a significant advantage in choosing DGC-MACS between cohorts of SDF with a 20% cutoff 325 [30].”

Author Response

Respected Editor and Reviewer

I am highly thankful and obliged to the respected reviewers and editors for the valuable time that you invested in re-evaluating the paper and thanks for the valuable comments you have provided. We have addressed all the comments raised by reviewers. Following are the answers

Reviewer 4

Overall, the manuscript needs English improvement. The manuscript must be carefully revised by the authors as it presents some mistakes and typographical errors that need to be corrected.

Response: Grammatical and typographical errors have been removed

Other than that, there are other points that need author’s attention and these are listed below.

Abbreviations should be explained within parentheses at their first mention in the manuscript.

Introduction, L 20/21, “However, the current success rates of ART remain subnormal”: What does it mean “subnormal”? Did you mean "relatively low" success rates?

Response: Changes made as per suggestions and rewritten the sentence

“However, the current success rates of assisted reproductive technology (ART) remain relatively low”.

Materials and Methods section needs to be reorganized and duplicities should be omitted (L 95/96 and L 115/116, L 94/95, L 109/111 and L 116, etc.).

Response: Omitted from the text as per kind suggestion

The study design is not clear. If the semen analyses were performed to select samples/participants for semen preparation methods, that should be clearly mentioned. Add "Sample collection and preparation" section.

Response: Done as suggested. Figure S1 has been added to make the design clear

2.4. Semen collection and Analysis

All men produced their semen sample in a sterile labelled jar, the semen sample was collected by masturbation on the day of oocyte aspiration, after 2–5 days of abstinence, and the collected semen sample was left to liquefy at 37°C for 30 min before analysis. Each sample was split into two aliquots: one was subjected to analysis for seminal characteristics.

Study design should include study subjects to be investigated (Inclusion/Exclusion criteria) with number of participants, Ethical issue, study groups.

Response: figure S1 has been added to make the design clear

2.2. Participants: First, male participants should be described, than their female partners. This section needs some revision to be more clear. For example, L95/96, “no infertility factor in the couple’s partner were included in the study”. What does it mean? Does it mean that male factor is a (confirmed)  contributing cause of the couple infertility? L 99/100: some of the andrological disorders are mentioned in the L 97.

Response: Excluded and revision made to make it clearer

2.3. This section is not clear. Is it related to study power calculation? L 106-111: this should be moved elsewhere. 

Response: Removed

2.4. Please explain the choice of 35 days of abstinence? Isn’t it too long (usually (and in WHO laboratory manual) it is suggested to abstain for 2-7 days before human semen sample collection). 

Response: It was typing error now changed to correct duration

“after 2–5 days of abstinence,”

2.4.1. In how many total spermatozoa cells counting was performed (to calculate the mean percentage)?

Response: 200 sperm added in text

“200 spermatozoa were observed with a phase contrast microscope, and the percentage of spermatozoa with tail changes typical of a reaction in the HOS test (swollen, HOS-reactive, or HOS-positive spermatozoa) was determined”

L 121-124: It is not clear – split the sentence.

Response: Rewritten to make it clear and splitting done 

2.4.2. ROS: Please, add the units in which the ROS value is expressed? What was the control? 

Response: Included in the method section

2.4.4. How is the semen thin film prepared?

Response: Included

Who is producer of CMA3 staining?

Response: Included and added in the text

What is a model of a fluorescence microscope? Please, provide Brand Name, Place of manufacture (City, Country).   

Included in the method section

Table 1: Comparative data for pre-pregnancy female BMI is missing. Statistical test used for data comparison should be added. Units for Anti Mullerian hormone levels, Total Gonadotropin dose per oocyte and Estradiol level are missing and should be added. 

Included in the table 1

Table 2, sperm concentration: What does it mean mx10exp6? Sperm count (total number of sperm in a sample of one ejaculate) is usually in the ranges of millions (and should be expressed as (UPPERCASE) M or 10exp6/ejaculate), and sperm concentration refers to the number of sperm per unit of volume (mL) of semen and is expressed in millions per mL (M/mL or 10exp6/mL).

Thanks for the clarification and suggestion Changes made to make it M/ml

 ROS%: what is used for the control? 

Included the control and the units

Discussion, 3.4. L 266, “a cohort of SDF with < 20 and ≥ 20%”, L 270 “cohort with SDF levels ≤ 20 %”, L 272 “in SDF ≥ 20 group”: If one group has SDF ≥ 20%, another group should have SDF < 20. 

Changes made already

L 268/269, “This improvement was more after DGC-MACS (z=-4.92, p<0.00) and (z=-6.4, p<0.00).”: It is not clear (for which group is first z and p, for which is the second). Please, revise all section to be clear what is significantly higher/lower in comparison to what.   

Rewritten the sentence to make it clear and understandable

Figure 1: If a difference in CMA3 % for some selection methods between the SDF groups is statistically significant (as is mentioned in the text (section 3.4.), please add * above the corresponding Box & Whisker. Explanation for Boxes and Whisker levels is missing (How are data spread out? What represents boxes and lines, what whiskers?). Unit for CMA3 is missing. 

Figure 1, Figure 2: Both figures show CMA3% levels for two SDF groups depending on the semen preparation methods. It is not clear. 

Figure one excluded to make the data more clear to infer

Discussion: It is not clear why is reference cited “We have not observed a significant advantage in choosing DGC-MACS between cohorts of SDF with a 20% cutoff 325 [30].”

Changed as per suggestion

“In the present study, we divided the male groups into a normal SDF and higher SDF range cohort to assess the true benefit of DGC-MACS. We have not observed a significant advantage in choosing DGC-MACS between cohorts of SDF with a 20% cutoff and similar findings were reported before”

Round 2

Reviewer 1 Report

I thank the authors for the revised version of their manuscript which has been improved. However, there are still problems with the methodology and interpretation of results that should be fully addressed.

Specific comments:

-It is insufficient to say that “female infertility was ruled out”. It is important to indicate what type of female factors were excluded.

-Flow cytometry is more sensitive than determining fluorescence using confocal microscopy because it counts at least 20,000 events and can also distinguish different intensities than confocal microscopy. Counting 100 spermatozoa is not comparable with the large number of events that quickly (within minutes) can be assessed with flow cytometry. If you want to use confocal microscopy because limitation of accessing a flow cytometry, then you need to count at least 200 cells in duplicate and need to determine the value of relative intensities by using ImageJ or other software that is capable of quantifying fluorescence.

“Actually, we counted 500 cells but at least 100 for each sample” This is not clear. What do you mean by counting 500 but then considering only 100 cells per sample? This is not enough.

-By confocal microscopy and using your eye, you are incapable of determining differences in fluorescence levels when comparing different spermatozoa. Using the procedure described above will help you to make this data meaningful.

- In this study, all procedures to select spermatozoa give similar results, then, it is not possible to conclude that DGC-MACS is a better technique to be used in the clinic. Moreover, DGC-MACS is not a novel technique as it has been used now for many years.

-There are no studies on apoptotic markers, and it is not clear whether the levels of ROS measured are high, so oxidative stress is present, or are normal levels for the technique used. Therefore, there is no indication of the presence of oxidative stress. Proper control values should be provided in the table.

Author Response

Respected Editor and reviewer

We are pleased to submit the revised version of the manuscript. We have addressed all the comments raised by the reviewers. Following are the answers to the comments

Reviewer1

Specific comments:

-It is insufficient to say that “female infertility was ruled out”. It is important to indicate what type of female factors were excluded.

Response: The female partner age < 35 years, female BMI < 24.5-18 kg/m2, FSH ≤10 IU/L (Day 2nd of menstrual cycle) AMH ≥ 1.0 ng/mL, OR antral follicle count >10, Evidence of at least one patent fallopian tube as determined by an hysterosalpingogram or laparoscopy showing at least one patent fallopian tube or a saline infusion sonogram showing spillage of contrast material, regular cycles defined as ≥25 days and ≤35 days in duration, Evidence of ovulation including biphasic basal body temperatures, positive ovulation predictor kits, or progesterone level ≥3 ng/ml,

-Flow cytometry is more sensitive than determining fluorescence using confocal microscopy because it counts at least 20,000 events and can also distinguish different intensities than confocal microscopy. Counting 100 spermatozoa is not comparable with the large number of events that quickly (within minutes) can be assessed with flow cytometry. If you want to use confocal microscopy because limitation of accessing a flow cytometry**** then you need to count at least 200 cells in duplicate and need to determine the value of relative intensities by using ImageJ or other software that is capable of quantifying fluorescence.

Response: We find your suggestions very innovative and interesting idea to use ImageJ to determine the intensities we will definitely look into it and incorporate it in our next research but the present method we followed the protocol as described earlier [1] which was based on spermatozoa exhibiting a bright green fluorescence in the head were scored as CMA3-positive. The technique we selection with confocal was due to limitation of accessing to flow cytometry  and we do agree that flow cytometry is more sensitive than determining fluorescence using confocal microscopy because it counts at least 20,000 events and can also distinguish different intensities but choosing confocal was due to test simplicity and easy to perform, inexpensive, and does not require sophisticated instrumentation for its evaluation, enabling its use in virtually all clinical laboratories.

 “Actually, we counted 500 cells but at least 100 for each sample” This is not clear. What do you mean by counting 500 but then considering only 100 cells per sample? This is not enough.

Response: We counted 500 spermatozoa / sample and statement is replaced in the sentence as per suggestion, ** {“200 spermatozoa were counted for each sample in duplication; bright yellow fluorescence reacted spermatozoa (CMA3+%) were considered chromatin uncondensed,”}.

-By confocal microscopy and using your eye, you are incapable of determining differences in fluorescence levels when comparing different spermatozoa. Using the procedure described above will help you to make this data meaningful.

Response: As said we will definitely consider your suggestion but flow cytometry was not done in this study.

- In this study, all procedures to select spermatozoa give similar results, then, it is not possible to conclude that DGC-MACS is a better technique to be used in the clinic. Moreover, DGC-MACS is not a novel technique as it has been used now for many years.

Response: As per suggestion the irrelevant statements and remarks removed from the text

-There are no studies on apoptotic markers,

Response: As per kind suggestion Statement removed

and it is not clear whether the levels of ROS measured are high, so oxidative stress is present, or are normal levels for the technique used. Therefore, there is no indication of the presence of oxidative stress. Proper control values should be provided in the table.

Response: Respected reviewer as per suggestion table for control value added

Reviewer 2 Report

All my comments have been addressed accordingly

Author Response

Thanking you 

Reviewer 3 Report

The revision is fine I recommend accept

Author Response

Thanking you

Reviewer 4 Report

Unfortunately, the revised parts of the manuscript were not marked, which made difficult to find what the authors changed. 

It seems that the authors have addressed most of concerns. I have some additional concerns:

Section 2.3. Sample size calculation: This section can be deleted. 

The first and last sentence of this Section can be moved to section 2.2, Line 102 (Please, add reference after “as previously described”, for example: Suresh, K., Chandrashekara, S. Sample size estimation and power analysis for clinical research studies. J Hum Reprod Sci, 2012, 5(1):7-13. doi: 10.4103/0974-1208.97779. Retraction in: J Hum Reprod Sci. 2015 Jul-Sep;8(3):186. PMID: 22870008; PMCID: PMC3409926).

Sentence in Line 114-119 (Simple randomization using a closed envelop method … cell selection (DGC-MACS), n=94.) can be incorporated in the Section 2.5. Experimental Design.

Line 129: Please, provide name and producer of Diff-Quik kit.

Table 1: Units for Total Gonadotropin dose and Estradiol level are missing and should be added.

Table 2 and 3, “HOS % (Hypo osmatic test)” should be replaced by “HOS % (Hypo-osmotic swelling test)”

Results, sections 3.3. and 3.4.: line 284 – add %, it should be “The CMA3+ % …” or add % after “21.9±7”, “27.5±10”  and “7.3±10.4”.  Same for the lines 300-303.  

Author Response

Respected Editor and reviewer

We are pleased to submit the revised version of the manuscript. We have addressed all the comments raised by the reviewers. Following are the answers of the comments

Reviewer 4

 Unfortunately, the revised parts of the manuscript were not marked, which made difficult to find what the authors changed. 

Response: Text Color changed to make it easier to fine

It seems that the authors have addressed most of concerns. I have some additional concerns:

Thanks for encouragement

Section 2.3. Sample size calculation: This section can be deleted. 

The first and last sentence of this Section can be moved to section 2.2, Line 102 (Please, add reference after “as previously described”, for example: Suresh, K., Chandrashekara, S. Sample size estimation and power analysis for clinical research studies. J Hum Reprod Sci, 2012, 5(1):7-13. doi: 10.4103/0974-1208.97779. Retraction in: J Hum Reprod Sci. 2015 Jul-Sep;8(3):186. PMID: 22870008; PMCID: PMC3409926).

Response: Deleted the said section as per recommendation and all said changes made as per suggestion

Sentence in Line 114-119 (Simple randomization using a closed envelop method … cell selection (DGC-MACS), n=94.) can be incorporated in the Section 2.5. Experimental Design.

Response: Section 2.5. Experimental Design said changes made

Line 129: Please, provide name and producer of Diff-Quik kit.

Response: Added in method section as per recommended

“Diff-Quik stain (Dade Behring Inc., Newark, NJ, USA).”

 Response: Added

Table 1: Units for Total Gonadotropin dose and Estradiol level are missing and should be added.

Response: Both parameters units added in table 1.

Table 2 and 3, “HOS % (Hypo osmatic test)” should be replaced by “HOS % (Hypo-osmotic swelling test)”

Response: Amendment done as per kind suggestion;

Table 2. Sperm concentration, volume, WBC/HPG, TNMS (total normal motile sperms), Normal % (normal morphology), HOS % (Hypo-osmotic swellin), SDF% (sperm DNA fragmentation), and CMA3% levels in the whole studied population”

Table 3. TNMS (total normal motile sperms),), HOS % (Hypo-osmotic swelling), SDF% (sperm DNA fragmentation), and CMA3+% levels in the whole studied population after preparation of patients with TZs.”

Results, sections 3.3. and 3.4.: line 284 – add %, it should be “The CMA3+ % …” or add % after “21.9±7”, “27.5±10”  and “7.3±10.4”.  Same for the lines 300-303.  

Response: Added as per kind input at the end of CMA3 %

  1. Lolis, D., I. Georgiou, M. Syrrou, K. Zikopoulos, M. Konstantelli, and I. Messinis, Chromomycin A3-staining as an indicator of protamine deficiency and fertilization. Int J Androl, 1996. 19(1): p. 23-7.

Round 3

Reviewer 1 Report

Thanks for answering to my comments and make improve the manuscript.